# Acute Coronary Syndrome: Disparities of Pathophysiology and Mortality with and without Peripheral Artery Disease

**DOI:** 10.3390/jpm13060944

**Published:** 2023-06-02

**Authors:** Flavius-Alexandru Gherasie, Mihaela-Roxana Popescu, Daniela Bartos

**Affiliations:** 1Department of Cardiology, University of Medicine and Pharmacy “Carol Davila,” 050474 Bucharest, Romania; 2Department of Cardiology, Elias Emergency University Hospital, Carol Davila University of Medicine and Pharmacy, 011461 Bucharest, Romania; 3Department of Internal Medicine, Clinical University Emergency Hospital, 014461 Bucharest, Romania

**Keywords:** peripheral artery disease, chronic limb-threatening ischemia, acute limb ischemia, ankle-brachial index, coronary artery disease, acute coronary syndrome, atherosclerosis, inflammation, plaque erosion, nodule calcification

## Abstract

There are a number of devastating complications associated with peripheral artery disease, including limb amputations and acute limb ischemia. Despite the overlap, atherosclerotic diseases have distinct causes that need to be differentiated and managed appropriately. In coronary atherosclerosis, thrombosis is often precipitated by rupture or erosion of fibrous caps around atheromatous plaques, which leads to acute coronary syndrome. Regardless of the extent of atherosclerosis, peripheral artery disease manifests itself as thrombosis. Two-thirds of patients with acute limb ischemia have thrombi associated with insignificant atherosclerosis. A local thrombogenic or remotely embolic basis of critical limb ischemia may be explained by obliterative thrombi in peripheral arteries of patients without coronary artery-like lesions. Studies showed that thrombosis of the above-knee arteries was more commonly due to calcified nodules, which are the least common cause of luminal thrombosis associated with acute coronary events in patients with acute coronary syndrome. Cardiovascular mortality was higher in peripheral artery disease without myocardial infarction/stroke than in myocardial infarction/stroke without peripheral artery disease. The aim of this paper is to gather published data regarding the disparities of acute coronary syndrome with and without peripheral artery disease in terms of pathophysiology and mortality.

## 1. Introduction

More than 200 million adults suffer from peripheral artery disease in their lower extremities, which increases their risk of cardiovascular events (such as coronary heart disease, strokes, and leg amputations). Globally and in the United States, peripheral artery disease has gone underdiagnosed and undertreated due to a lack of awareness [1]. Generally speaking, lower-extremity peripheral artery disease refers to atherosclerotic diseases of the arteries supplying the limbs, from the aortoiliac segments to the pedal arteries. Even though this disease is associated with adverse clinical outcomes, impaired physical function, and reduced physical activity, it has been understudied and underrecognized compared with other atherosclerotic diseases such as myocardial infarction. In recent years, there has been mounting evidence that peripheral artery disease is significantly linked to mortality, primarily as a risk factor for future myocardial infarctions and strokes. Also, peripheral artery disease can cause devastating complications that result in limb amputations and acute limb ischemia. Despite the overlap, the causes of atherosclerotic diseases are not the same and the need for the appropriate diagnosis and treatment remains a major concern for medicine. Clinically, peripheral artery disease and coronary artery disease overlap due to their shared risk factors. In these patients, hyperlipidemia and type II diabetes mellitus were significant comorbidities. In addition to accelerating atherosclerosis development, diabetes mellitus affects the characteristic of atherosclerotic lesions in the lower extremities [2,3]. Compared with patients with PAD (peripheral artery disease) and without diabetes mellitus, patients who have PAD and coexisting diabetes mellitus have more arteries below the knee that are affected and more multilevel lesions [4,5,6]. Although patients with diabetes mellitus benefit from medical and endovascular procedures to manage atherosclerotic cardiovascular diseases, restenosis is a significant factor that may worsen the prognosis and may be associated with the necessity of reintervention [7]. The presence of dyslipidemia is associated with both changes in basic lipid parameters and modified lipoproteins, which are not routinely measured in clinical practice. Nitrated lipoproteins are oxidatively modified lipoproteins discussed in the context of cardiovascular dysfunction development. However, their role remains unclear at this time [8]. Many studies have found that coronary artery disease and peripheral artery disease are frequently coexisting conditions. Saleh et al. suggested that there was a significant increase in peripheral artery disease prevalence among those with coronary artery disease compared with those with normal coronaries [9]. Despite being under the care of a cardiovascular specialist, studies have shown that one out of six patients with coronary artery disease had an unrecognized peripheral artery disease [10].

The ankle-brachial index’s high positive predictive value and the high prevalence of peripheral artery disease with coronary artery disease suggest a high degree of suspicion for, and pretest probability for, coronary artery disease in a patient with peripheral artery disease. Coronary artery disease diagnosis is highly dependent on cardiac imaging tests such as electrocardiograms, stress electrocardiograms, myocardial perfusion imaging, and coronary angiographies. Kumar et al. say pretest probabilities can be increased through the ankle-brachial index, but it cannot replace the above testing methods [11].

Many factors contribute to this lack of knowledge about peripheral artery disease. Various nomenclatures and definitions have been used to describe peripheral artery disease, making effective communication challenging [12]. It is also likely that the clinical presentation of peripheral artery disease contributes to confusion regarding peripheral artery disease. A small percentage of patients are diagnosed with intermittent claudication, with either no exertional leg symptoms (up to 50%) or atypical leg symptoms (almost 50%) [13]. Patients may experience leg pain that begins at rest, or that does not interfere with walking, or that resolves with rest. These symptoms can be confused with arthritis or degenerative spinal disease. Various adverse outcomes can occur as a result of peripheral artery disease. To improve overall outcomes among this growing and undertreated population, increasing awareness about the definition, diagnosis, clinical manifestations, and complications of peripheral artery disease is critical. 

Coronary atherosclerosis often manifests as an acute coronary syndrome, where thrombosis is precipitated by rupture or erosion of the fibrous caps of atheromatous plaques. It is common for plaque ruptures to have large necrotic cores, as well as thin, inflamed fibrous caps. Regardless of the extent of atherosclerosis, peripheral artery disease manifests as thrombosis. In peripheral arteries with significant stenosis, approximately 75% of them are blocked by thrombi. Two-thirds of them also have thrombi associated with insignificant atherosclerosis. A local thrombogenic or remotely embolic basis of critical limb ischemia may be explained by obliterative thrombi in peripheral arteries of patients without coronary artery-like lesions [14].

This study aims to study the prevalence of peripheral arterial disease and its correlation with the severity and characteristics of patients’ coronary disease, as well as its impact on in-hospital mortality and major adverse cardiac. The prevalence of peripheral arterial disease varies in the studies carried out so far.

## 2. Anatomy of Coronary Arteries: Normal and Pathological Aspects

The epicardial coronary artery systems are the left and right coronary arteries, which normally arise from the left and right ostia, located in the sinuses of Valsalva [15,16]. A separate ostium in the right sinus gives rise to the “third coronary artery” in about half of humans [17]. Several smaller ostia can be found in the right sinus, giving rise to multiple right ventricular branches [18,19].

The right coronary artery travels down the right atrioventricular groove, toward the crux of the heart. In addition to supplying blood to the right ventricle, the right coronary artery supplies 25% to 35% of the left ventricle. The main branches of the right coronary artery are the following: the conus artery, the sinoatrial nodal artery, the acute marginal artery, and the posterior descending artery [20].

The left main coronary artery has an overall mean length of 10.5 ± 4 mm for men and 10 ± 3 mm for women, it enters the coronary sulcus after passing between the main pulmonary artery and the left auricle, and bifurcates into the left anterior descending artery and the left circumflex [21].

From the bifurcation of the left main coronary artery, the left anterior descending artery continues around the left side of the pulmonary artery and descends in the epicardial fat into the anterior interventricular sulcus of the heart; its normal length is from 10 cm to 13 cm [22]. There are three segments of the left anterior descendent artery: a proximal segment (from the origin to the first septal artery), a middle segment (from the first septal to the origin of the first diagonal branch), and a distal segment (from the origin of the first diagonal artery to the apex). There are a number of diagonal branches of the left anterior descending artery, one or two large branches that are usually present along the anterior surface of the left ventricle, and the septal perforator branches, which supply the anterior 2/3 of the basal interventricular septum and the entire septum at its mid and apical levels [20,22,23].

From the left main coronary artery bifurcation, the left circumflex artery traverses the coronary sulcus and the diaphragmatic surface before terminating at the posterior interventricular sulcus. In contrast with the right coronary artery and left anterior descending arteries, the circumflex artery has only two segments, called the proximal segment and distal segment [24]. Its normal length is from 5 cm to 8 cm. There are several branches that originate from the left circumflex artery, including the left marginal artery, the posterolateral branch, and the obtuse marginal branch, which supplies a portion of the inferior wall [20,23].

## 3. Anatomy of Lower Extremity Arteries: Normal Aspects

To maintain the body’s mobility, lower extremity arteries supply oxygenated blood to muscles, tendons, and nerves. Several diseases can affect these arteries, which inhibit the optimal function of the limb. Lower-extremity arteries originate from the iliac artery at the point where the abdominal aorta is divided into the two common iliac arteries and the median sacral artery, approximately at the fourth lumbar vertebral body [25,26].

There are two branches of the common iliac artery, the internal and the external. A portion of the external iliac artery runs down into the lower limbs and becomes the common femoral artery [27]. Among the small branches of the common femoral artery, there are the superficial epigastric artery, external pudendal artery, and superficial circumflex artery. The common femoral artery divides into the superficial and deep femoral arteries. 

Through Hunter’s canal, the superficial femoral artery travels along the medial side of the thigh. Once it exits the adductor canal, the superficial femoral artery courses posteriorly, where it is called the popliteal artery after it passes through the adductor hiatus. Approximately at the level of the proximal tibiofibular joint, the popliteal artery divides into the anterior tibial artery and the tibioperoneal trunk [28]. The anterior tibial artery runs along the anterior surface of the interosseus membrane, and there is a continuation of this artery in the foot as the dorsalis pedis artery. Located along the posteromedial aspect of the leg, the tibioperoneal trunk further divides into the peroneal artery and posterior tibial artery [25]. There are two arches on the medial and lateral surfaces of the plantar surface, formed by the posterior tibial artery. Besides giving rise to the metatarsals, the plantar arch is also responsible for producing the plantar digital arteries. Above the ankle joint, the peroneal artery divides into two calcaneal branches, one on the medial side and one on the lateral side. As a result of free communication with the dorsalis pedis artery and posterior tibial artery, these branches assist in the collateralization of the foot when it is sick or injured [29].

## 4. Atherosclerosis

Atherosclerosis is a chronic inflammatory disease with complex etiopathogenesis, which results in the development of atherosclerotic plaques, leading to the formation of narrowing and/or occlusion of the arteries [30,31]. This clinically may lead to the development of ischemic heart disease, cerebrovascular disease, or peripheral arterial disease [32,33,34,35]. In clinical practice, lowering the level of LDL cholesterol (low-density lipoprotein cholesterol) is the primary therapeutic objective of lipid-lowering therapy due to the role that oxidatively modified LDL particles play in atherosclerosis [36,37,38]. Additionally, we observe patients with advanced atherosclerosis and normal LDL concentrations in clinical practice, and such patients are also eligible for statin therapy. There are many pleiotropic effects associated with statins, which are mainly used for lowering cholesterol, and thus they are not only lipid-lowering drugs [37]. To achieve a target level for LDL cholesterol, statins and PCSK9 inhibitors (proprotein convertase subtilisin/kexin type 9) are effective [39,40,41]. There is no doubt that both of them are effective in controlling LDL cholesterol and reducing major adverse cardiovascular events by about 50% [42]. After controlling LDL cholesterol, the remaining risk for major adverse cardiovascular events is believed to be due to inflammation. As part of the CANTOS trial, it was shown that canakinumab, an antibody that blocks IL-1β (interleukin-1β), reduces major adverse cardiovascular events [43,44,45,45,46,47].

As a result of atherosclerosis, there is an autoimmune response to LDL and other antigens which could lead to an escalation or amelioration of the disease’s progression. There have been some recent advances in immunotherapy and vaccination that have shown promise in curbing atherosclerosis in animal models [48]. Based on the theory of the modulation of atherogenesis by adaptive immune responses, especially the CD4+ T cells that recognize self-antigens, a novel type of therapy that targets the adaptive immune response has been developed to address this phenomenon [49].

### 4.1. Pathogenesis of Atherosclerosis 

Early-stage atherosclerosis is characterized by cell formation which is the result of lipid accumulation in the cells [50,51,52]. Inflammatory cytokines are released by damaged endothelial cells, which recruit monocytes through the endothelium and then differentiate into macrophages. Differentiated macrophages consume oxidized low-density lipoprotein (ox-LDL) through the scavenger receptors LOX-1, CD36, and SR-A1, which hydrolyze cholesteryl esters within lysosomes. By contrast, free cholesterol is esterified by acyl-CoA cholesterol acyltransferase 1 (ACAT1), while cholesteryl ester is hydrolyzed by neutral cholesteryl ester hydrolase (nCEH) and cholesteryl ester hydrolase (CEH) [51,53]. A large amount of free cholesterol is toxic to cells, so it must be efficiently removed from cells by ABCA1 and ABCG1 transporters [54,55]. When the above cholesterol homeostasis is disturbed, excessive cholesterol ester or free cholesterol will accumulate, causing foam cell formation or cell necrosis.

### 4.2. Immune Response in Atherosclerosis

Atherosclerotic plaques can become unstable, rupture, or erode over time, causing major cardiovascular complications [55,56,57,58,59]. There is a direct correlation between plaque stability and the level of inflammatory cells as well as the thickness of the cap of the plaque. Plaques that have thin caps and are full of immune cells are referred to as soft or vulnerable plaques. To initiate immune cell infiltration, chemokines and adhesion molecules play a major role. An antigen-presenting cell is a cell that generates major histocompatibility complex molecules, costimulatory molecules, and cytokines upon the presence of antigens that are produced by pathogens, bacteria, or altered self to determine the polarization of the adaptive immune response. 

In the arterial adventitia and neointima, there are macrophages and dendritic cells that are activated by TLR ligands (Toll-like receptors) and scavenger receptors [60]. As inflammation cytokines increase in amount and intensity, atherosclerosis gets worse, and more immune cells are attracted to the area. The inflammatory cytokine IL-1β is an effective target for the treatment of atherosclerosis and other vascular diseases [58]. 

Marchini et al. suggested that at the maturation stage of atherosclerotic plaque, atherosclerosis-antigen-specific T cells release cytokines, perpetuate inflammation, and help an immune cell infiltrate develop over time [61] (Figure 1).

There is always an autoimmune response associated with atherosclerosis. There are antibodies to oxidized (oxLDL) LDL that are produced by plasma cells derived from the B cell, and these antibodies can be detected in the serum of humans and animals suffering from atherosclerosis. The presence of T cells against antigens associated with atherosclerosis is also found. For some time now, regulating CD4+ T cells (Tregs) have been shown to protect mice from atherosclerosis in animal models [63]. Our recent studies showed that CD4+ T cells specific for the core lipoprotein ApoB (apolipoprotein B), which is involved in LDL, very low-density lipoproteins, and chylomicron, are mostly Tregs in people without cardiovascular disease, but assume mixed and effector phenotypes in those with cardiovascular disease [49,64]. There is some evidence that atheroprotection may involve other T- and B-cell subsets, and this is an area of active research. 

### 4.3. Atherosclerosis and Autoreactive CD4+ T cells

A majority of CD4+ T cells present in atherosclerotic lesions are memory T cells (CD45RO+) and are responsible for producing inflammatory cytokines in response to oxLDL. Several immunogenic epitopes have been identified and used as vaccine antigens against atherosclerosis in animal models derived from mouse ApoB (Apolipoprotein B).

Until recently, there has not been any direct proof that CD4+ T cells specific for ApoB epitopes exist in the body. To answer this question, scientists developed MHC-II tetramers for detecting such cells in mice as well as humans. ApoB epitope P18 is identical in mice (ApoB) and humans (APOB). P18 binds the mouse MHC-II allele I-Ab and DRB1*07:01 (presented in about 8% of humans). To detect APOB-specific CD4+ T cells, scientists created human APOB-peptide P18:DRB1*07:01 tetramers and found that P18-recognizing CD4+ T cells exist in human peripheral blood mononuclear cells. The presence of these cells was found in subjects both with and without subclinical cardiovascular disease [49]. The results of this study are the first in a series of studies to demonstrate that there are self-peptide-recognizing CD4+ T cells within human peripheral blood mononuclear cells. During the progression of atherosclerosis, the phenotypes of these cells seem to change [64,65,66,67,68,69].

### 4.4. Future Atherosclerosis Prevention

There is no doubt that vaccination is one of the most successful interventions in medicine. In recent years, vaccine development has moved from vaccine development for infectious diseases to vaccine development for non-communicable diseases, such as cancer, atherosclerosis, hypertension, Alzheimer’s disease, and diabetes mellitus. Identifying vaccine antigens is the first step in developing an atherosclerosis vaccine. PCSK9 (proprotein convertase subtilisin/kexin type 9), HSP65, and ApoB are some of the possible antigens that may be used as atherosclerosis vaccine antigens [70]. There are already antibodies against PCSK9 that are being used in clinical trials. As far as targeting PCSK9 is concerned, it is known to be safe because humans with null mutations in PCSK9 are asymptomatic except for the fact that they are resistant to atherosclerosis [71]. 

There is currently no information available regarding how long vaccination-induced atheroprotection lasts, how often the vaccine needs to be administered, and what would be the most effective dose, formulation, and route of administration for this vaccine.

## 5. Diagnostic Approach to Coronary Artery Disease

The first step is to evaluate the patient’s general condition and quality of life by assessing the comorbidities, symptoms, and signs. While physical examination findings generally do not indicate that acute coronary syndromes are present, thorough patient evaluation is essential for assessing immediate risk, recognizing hemodynamic collapse, and identifying mechanical complications associated with myocardial infarction. The presence of a tachycardia, narrow pulse pressure, hypotension, and signs of congestion (for example, pulmonary oedema) or inadequate perfusion (for example, cool extremities) are all indicators of a high clinical risk. With the Killip classification, patients with acute coronary syndromes are classified according to the degree of clinical heart failure, ranging from no signs of congestive heart failure (Class I) to cardiogenic shock (Class IV), and this classification is highly predictive of mortality [72,73]. Typically, mechanical complications from myocardial infarction are accompanied by abrupt haemodynamic deterioration and a loud holosystolic murmur in the left parasternal region in cases of acute ventricular septal rupture, soft systolic murmurs in cases of acute mitral regurgitation, and signs of tamponade when free wall rupture occurs [74].

The diagnosis of acute coronary syndrome is based on clinical presentation, ECG (electrocardiogram) findings, and biochemical evidence of myocardial injury. In the immediate context of a patient suspected of having acute coronary syndrome, it is important to rule out the presence or absence of ST-segment elevations on a 12-lead electrocardiogram [75,76]. 

Since the development of high-sensitivity troponin (hsTn), the diagnosis of NSTEACS (non-ST-elevation acute coronary syndromes), which include unstable angina and NSTEMI (non-ST segment elevation myocardial infarction), has advanced significantly. NSTEACS may present with T-wave inversions or ST-segment depressions on the ECG, but these findings are not necessary for diagnosis. NSTEMI is distinguished from unstable angina by elevated levels of myocardial necrosis markers such as cardiac troponin I and T (cTnI and cTnT) or creatine kinase-myocardial band (CK-MB). This is usually accompanied by an initial rise and a peak, followed by a fall in the concentration of the biomarker [77,78].

The most recent European and American guidelines emphasize the importance of emergent reperfusion therapy for patients who manifest ST-segment elevation infarction [76,79]. If a patient presents to a hospital that has the capability of performing percutaneous coronary intervention (PCI), immediate coronary angiography should be performed with a goal of limiting the time from first medical contact to the insertion of the device to less than 60–90 min. Unless PCI is possible within this timeframe, fibrinolysis should be administered if not contraindicated, and a pharmaco-invasive approach should be considered, in which fibrinolytic therapy is followed by invasive angiography within 24 h [76,79]. 

In regard to NSTEACS, the optimal time for catheterization continues to be debated. In the most critical cases (such as those with hemodynamic instability, chest discomfort that persists or arrhythmia that poses a life-threatening danger), emergent angiography should be performed within two hours of hospital admission. Those with a high risk of coronary heart disease (e.g., GRACE scores >140) will undergo catheterization within 24 h, while those with a low risk will undergo selective procedures [76].

The electrocardiogram and the assessment of the function of the left ventricle are included in the basic testing. All patients should undergo a resting transthoracic echocardiogram to rule out alternative causes of angina, to detect regional wall motion abnormalities indicative of coronary artery disease, to assess left ventricle ejection fraction for risk-stratification purposes, and to assess diastolic function [80].

In patients with suspected or established CCS (chronic coronary syndrome), the following clinical scenarios are most frequently encountered: patients with suspected CAD (coronary artery disease) and stable anginal symptoms and/or dyspnoea; patients with new onset of heart failure or left ventricular dysfunction and suspected CAD; asymptomatic and symptomatic patients with stabilized symptoms < 1 year after an ACS; patients with recent revascularization; asymptomatic and symptomatic patients > 1 year after initial diagnosis or revascularization; patients with angina and suspected vasospastic or microvascular disease; and asymptomatic subjects in whom CAD is detected at screening [1]. 

Based on recent meta-analyses of the performance of diagnostic tests for the detection of anatomically (>50%) and functionally significant coronary artery disease, there is different sensitivity and specificity between computed tomography angiography, positron emission tomography, single-photon emission computed tomography (exercise with or without dipyridamole or adenosine), stress cardiac magnetic resonance, stress echo exercise, stress echocardiography, and invasive angiography [80]. 

The presence of peripheral artery disease (PAD) is an essential factor to consider when discussing the outcomes of percutaneous coronary intervention (PCI) and coronary artery bypass grafting (CABG) for patients with acute coronary syndrome (ACS). Patients with PAD face unique challenges when undergoing these procedures, making outcomes assessment particularly significant.

The extent of PAD can significantly influence the treatment of ACS and can affect the outcomes of PCI and CABG. Patients with PAD may have a higher risk of adverse events following PCI, including mortality, major bleeding, and urgent revascularization [81]. Likewise, CABG may be associated with higher stroke rates and re-operation in PAD patients. As such, it is essential to consider the presence of PAD when assessing PCI and CABG outcomes for ACS patients [82]. This may involve additional risk stratification, pre-procedural testing, and post-procedural monitoring [83].

## 6. Diagnostic Approach to Peripheral Artery Disease

As a result of technological advances over the past decade, noninvasive imaging has become a very effective tool for assessing the anatomy and severity of arterial stenosis in patients with peripheral artery disease. The main advantages of this technology are its ability to image distal vessels with calcification, a lower contrast dose, and a higher spatial resolution, among others [84]. 

Patients with peripheral arterial disease often experience intermittent claudication. Intermittent claudication occurs when blood circulation to the muscles is blocked due to obstruction to arterial flow during exercise, resulting in pain in the calves and less commonly the hips and buttocks. There is a range of severity for symptoms, which is summarized on Fontaine’s scale (Table 1) and Rutherford’s scale (Table 2) [85,86].

This first-line noninvasive diagnostic method is based on the ratio of ankle-to-brachial systolic blood pressure, a measurement that requires standard measurement methodology. The presence of peripheral artery disease is determined by an ankle-to-brachial index ≤0.90 [87]. There are many cases of peripheral arterial stenosis even when the ABI is normal or elevated; in such cases the toe-brachial-index (TBI) should be used. To calculate TBI, it is commonly recommended to use the great toe artery pressure since the vessels of the toes are not usually affected by media sclerosis [88]. TBI’s utility in diabetic patients makes it the method of choice to evaluate lower-limb perfusion when there is overt arterial wall calcification. According to the algorithm devised by Park et al., wound management is recommended if the ABI is less than 0.9 or if the TBI is less than 0.6 after angiography. Further, if the TBI was less than 0.6 in patients with normal ABI values, angiography was performed prior to wound treatment. In patients with diabetic gangrene, this treatment algorithm will allow more rapid detection of PAD, improving wound management and reducing hospitalizations [89].

The definition of chronic limb-threatening ischemia (CLTI), the most severe form of peripheral arterial disease, should encompass a broader and more heterogeneous group of patients with varying degrees of ischemia that may delay wound healing and increase the risk of amputation. A clearer concept of CLTI needs to be developed by excluding the following individuals from the population as defined by the guideline document: patients with purely venous ulcers, acute limb ischemia (ALI), acute trash foot, ischemia caused by emboli, acute trauma, or mangled extremities, as well as those with wounds associated with non-atherosclerotic conditions [14,90,91]. In order to establish a diagnosis of CLTI, an objectively documented atherosclerotic PAD must be associated with ischemic rest pain or tissue loss (ulceration or gangrene) [92]. Ischemic rest pain should persist for at least two weeks and be associated with at least one abnormal hemodynamic parameter: (ABI) < 0.4, absolute highest AP (ankle pressure) < 50 mm Hg, absolute TP (posterior tibial artery) < 30 mm Hg, transcutaneous partial pressure of oxygen (TcPo2) < 30 mm Hg, or flat or minimally pulsatile pulse volume recording (PVR) waveforms. Since the 1950s, a number of different classification systems for lower-limb ischemia and wounds have been developed and promulgated [91]. Among the most widely accepted classifications are the Fontaine and Rutherford classifications [86].

Acute limb ischemia (ALI) refers to a rapid decrease in lower-limb blood flow caused by an acute occlusion of a peripheral artery or bypass graft, and the prognosis is poor unless the condition is treated quickly and appropriately. Embolizations and thromboses are the most common etiologies, with a variety of comorbidities [92]. The symptoms of ALI are abrupt and include pain, numbness, and coldness of the lower limbs, along with paresthesia, contracture, and irreversible purpura. There are several treatments available for ALI, including open surgical revascularization, endovascular revascularization, and heparin administration within a short period of time. As a result of the poor prognosis for limb recovery, limb amputation should be performed without hesitation if the limb is irreversible [93,94,95,96]. 

Duplex ultrasound, computer tomographic angiography, magnetic resonance angiography, and invasive angiography have a different sensibility and specificity for detecting peripheral artery disease. For example, when comparing multidetector computer tomography with angiography, the sensitivity and specificity of multidetector computer tomography for detecting peripheral artery disease is 90% [97,98,99].

However, to diagnose peripheral artery disease, catheter-based angiography remains the gold standard, although it is now limited to patients who are receiving endovascular revascularization as part of the process [100]. 

## 7. Pathology of Coronary Artery Disease and Acute Coronary Events

Approximately two-thirds of all acute coronary events are caused by the rupture of the thin-capped fibroatheromas in the plaque, while one-third are caused by plaque erosion caused by pathological intimal thickening or fibroatheroma in the plaque. The calcification of a nodule can sometimes penetrate beyond the fibrous cap and result in the formation of a thrombus. As a result of plaque rupture, a loss of integrity occurs in the fibrous cap, which is responsible for separating the necrotic core in the lumen of the artery from the blood. This defect is the primary cause of coronary thrombosis present in autopsies of patients who died unexpectedly [101,102]. During the rupture of the plaques, the necrotic cores constituted more than 30% of the plaque area. Several macrophages and a few smooth muscle cells were also present in the thick fibrous cap covering the plaques. There was a tendency for such plaques to be accompanied by spotty calcification, and the plaque cores often showed that hemorrhage had occurred.

Kolofgie et al. suggested that plaque hemorrhage is the result of leaky neovasculature caused by proliferating adventitial vessels, resulting in proliferating adventitial vasa vasorum, which further expands the plaque core, as the erythrocyte membranes are rich in free cholesterol [56]. 

There is a risk of large intraplaque hemorrhages due to neovessel ruptures. A plaque rupture can also result in luminal blood being able to reach the necrotic core of the plaque, contributing to the enlargement of the plaque, especially in the event of silent ruptures. There is intense inflammation within the disrupted plaques, which stretches from the fibrous cap to the necrosis and adventitious layer of the vessel wall [103]. As observed in this study, fibrous cap attenuation was one of the strongest predictors of high-risk plaques. It was found that stable plaques had a fibrous cap thickness of 85 microns, and disrupted and vulnerable plaques had a fibrous cap thickness of 55 microns [104,105]. Two other characteristics distinguished this type of plaque from others: the extent of inflammation within the plaque and the extent of necrosis within the plaque. Furthermore, the plaque was more likely to become unstable, as a result of a greater degree of luminal obstruction.

## 8. Pathology of Peripheral Artery Disease

In patients with coronary artery disease, sudden cardiac death and myocardial infarction are the most common adverse events that occur. When these situations occur, the underlying pathology is in situ thrombosis, which is primarily caused by rupture of the atherosclerotic plaque of the underlying atherosclerotic artery, and rarely by erosion of the plaque itself. In similar ways, acute limb ischemia is characterized by sudden decreases in perfusion in the limbs that can lead to tissue loss and may require early intervention to prevent further complications. 

Acute limb ischemia in peripheral artery disease is, however, very different from acute coronary events wherein the underlying pathology is atherothrombosis. This is due to in situ thrombosis, embolisms from the heart and proximal vessels, and graft occlusion as well [106]. In general, ALI progresses into advanced limb ischemia within two weeks of its acute onset, and it is associated with a mortality rate of 15–20% due to concurrent illnesses, such as cardiovascular disease and ischemia–reperfusion injury [107]. Acute limb ischemia, excluding trauma, is generally classified as embolism and thrombosis. A recent report found that embolisms, thromboses caused by occlusive atherosclerotic lesions, complex factors, and stent or graft-related thromboses occur at rates of 46%, 24%, 20%, and 10%, respectively [107]. It is not uncommon for ALIs to occur due to peripheral embolisms caused by aneurysms of the popliteal artery and thrombotic occlusion of aneurysms. 

Thrombosis occurs when chronic stenotic lesions in occlusive atherosclerosis break down, fail to circulate, or become hypercoagulable. There is often difficulty distinguishing embolism from thrombosis in the presence of stenotic arteriosclerosis obliterans lesions [107].

According to Narula et al., 239 arteries were pathologically examined in 75 patients with critical limb ischemia who had undergone 98 amputations above the knee and below the knee [108]. According to our study, one-fourth of the arteries with luminal stenosis of at least 70% had stenosis due to significant atherosclerosis without thrombi in the arteries. It has been reported that thrombi contributed to luminal compromise in two-thirds of patients, with only one-third of these having significant atherosclerosis associated with it and the remainder having nonsignificant atherosclerosis associated with it. It is important to note that luminal thrombi associated with significant atherosclerosis were considered in situ thrombi possibly related to plaque rupture, erosion, or calcified nodules, whereas luminal thrombi associated with insignificant atherosclerosis suggest that they may be associated with thromboembolism. There was medial calcification in more than 70% of all arteries. While it is suggested that luminal thrombi associated with insignificant atherosclerosis are likely thromboembolic, it has also been suggested that the non-compliant vessels with medial calcification could have also contributed to luminal thrombi caused by the stasis of blood owing to the lack of mobility, infection, and non-compliant vessels [91]. 

## 9. Pathological Comparison of Arteries of Patients with Coronary Artery Disease and Peripheral Artery Disease

There are several clinical implications related to the pathological changes in the lower extremity arteries of patients with peripheral artery disease, specifically those who suffer from major adverse limb events. Among the most serious adverse limb events occur when the limb is severely ischemic and thus requires intervention or amputation due to vascular damage. Occlusion of large arteries by thrombus without significant atherosclerosis and the occlusion of small subcutaneous arteries by thrombus suggest the possibility of thromboembolism. It has been observed that repeated, layered thrombi are present in several arteries that are free of significant atherosclerosis, thus supporting the idea that emboli are repeated over time [91,109,110,111,112,113]. To further understand the factors that contribute to the development of thrombotic lesions, in addition to alterations in the coagulation cascade, infection, diminished mobility, and non-compliant vessels, additional studies are required. Acute coronary syndrome, on the other hand, is commonly associated with in situ luminal thrombus caused by the rupture of the thin-capped fibroatheroma plaque [114,115,116].

Torii et al. studied 3000 serial autopsies and showed that thrombosis of the above-knee arteries was more commonly due to calcified nodules, which are the least common cause of luminal thrombosis associated with acute coronary events in patients with acute coronary syndrome [110,117]. There is a higher prevalence of fibrocalcific plaques in the lower extremities as compared with the coronary arteries [110,118]. In patients with peripheral artery disease, calcification of the middle layers of arterial walls is common, but it is rare in the coronary arteries unless the patient also has advanced kidney disease [119]. Several pathological factors could contribute to the failure of any artery revascularization procedure, such as rigid, non-expansive, noncompliant large and small arteries with medial calcification of the intermediate vessels and distal vessel thrombosis [120]. 

In conclusion, occlusive complications caused by atherosclerotic lesions can occur locally or distally, depending on the extent of the disease. A plaque rupture, plaque erosion, or calcium-rich nodules protruding through the plaque surface are generally responsible for local complications. There is no better example than what we see in the presence of coronary artery disease [121,122,123]. There is, however, the possibility of embolic compromise in distal territories, whereby aortic, iliac, and femoral plaques can impair infra-popliteal arterial supply and result in critical limb ischemia.

## 10. Mortality and Cardiovascular Outcomes

When it comes to mortality in patients with peripheral artery disease, The Ankle to Brachial Cohort Study found a strong association between low (0.90) and high (>1.40) ankle-to-brachial results and all-cause and cardiovascular mortality [124]. The mortality rate was doubled in those with an ankle-to-brachial index between 0.81 and 0.90, while the mortality rate was quadrupled in those with an ankle-to-brachial index between 0.70 and 0.80. There has been evidence in multiple studies from a variety of populations demonstrating that persons with peripheral artery disease are more likely to develop other cardiovascular diseases as well, such as coronary heart disease, strokes, and aneurysms of the abdominal aorta [125,126]. 

Recent research has shown that peripheral artery disease is becoming increasingly important in the context of polyvascular disease. An example of this is a subset of patients with multiple vascular beds affected by atherosclerosis, including peripheral arterial disease. According to the FOURIER trial, peripheral artery disease in combination with myocardial infarction or stroke had the highest risk of major adverse cardiovascular events (cardiovascular mortality, myocardial infarction, and stroke) over 2.5 years, with a risk of 14.9% [127,128,129,130,131] (Figure 2). 

There is an interesting finding in this study that peripheral artery disease without myocardial infarction/stroke was associated with a higher risk of major adverse cardiovascular events (10.3%) than myocardial infarction/stroke without peripheral artery disease (7.6%) [132,133,134,135,136]. 

Numerous studies have demonstrated a correlation between peripheral artery disease and increased mortality (Table 3) [127,137,138,139,140].

The mortality and adverse outcomes associated with acute coronary syndrome and peripheral artery disease are not only unfortunate consequences of the natural progression of these diseases but can also result from medical, endovascular, or surgical interventions [141]. While potentially beneficial, these interventions can have unintended consequences that may result in death or other adverse outcomes.

## 11. Conclusions

Overall mortality rates for acute coronary syndromes and acute limb ischemia have declined in most developed countries by 24–50%. It is estimated that approximately half of this reduction in cardiovascular mortality can be attributed to changes in therapy, including secondary preventive measures following myocardial infarction and revascularization. Initial treatments, advancements in heart failure treatments, and revascularization for chronic angina contributed to this drop. The other half of the effect was caused by changes in risk factors, including reductions in total cholesterol, systolic blood pressure, smoking, and physical inactivity. In addition, detecting peripheral arterial disease and developing novel treatments can improve our therapeutic options. 

Slowing the progression of peripheral artery disease is key to decreasing cardiovascular mortality. For now, lifestyle changes, dietary changes, and conventional therapies may seem like a winning combination, but vaccines might be the medicine of the future.

## Figures and Tables

**Figure 1 jpm-13-00944-f001:**
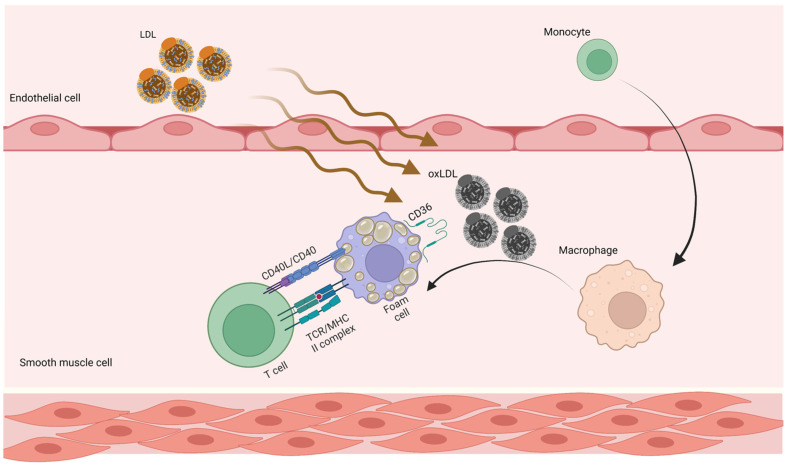
Adaptive immune responses in atherosclerosis. LDL penetrates the artery wall and experiences modification by oxidative and enzymatic processes. The modified LDL molecules promote the expression of leukocyte adhesion molecules. Monocytes invade the vascular wall and mature into macrophages, differentiating into foam cells after taking up large volumes of oxidized LDL. T cells become active throughout this process and release mediators, which subsequently increase the immune reaction and lead to atherogenesis. LDL, low-density lipoprotein; oxLDL, oxidized low-density lipoprotein [62].

**Figure 2 jpm-13-00944-f002:**
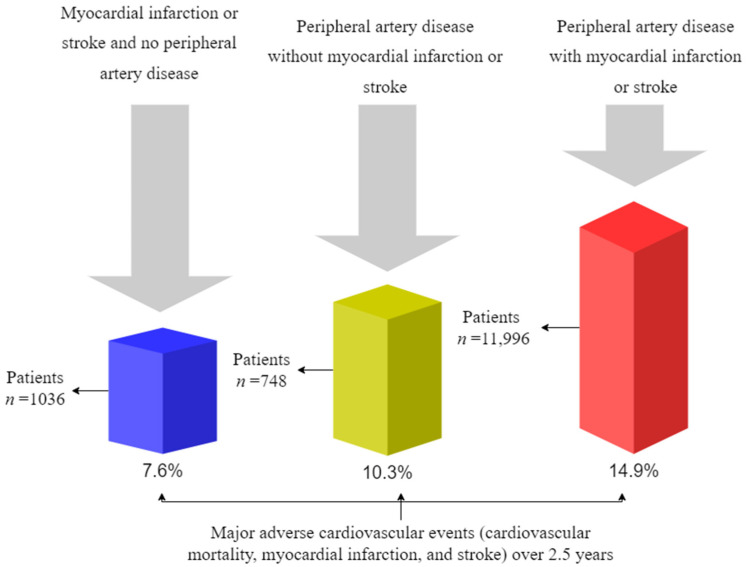
Cumulative incidence of major adverse cardiovascular events in the placebo group according to CVD status at baseline [127]. CVD indicates cardiovascular disease.

**Table 1 jpm-13-00944-t001:** The Fontaine classification.

Stage I	Asymptomatic
Stage IIa	Intermittent claudication after more than 200 m of walking
Stage IIb	Intermittent claudication after less than 200 m of walking
Stage III	Rest pain. Rest pain appears especially during the night when the legs are raised up on to the bed, which diminishes the gravitational effect present by day
Stage IV	Ischaemic ulcers or gangrene (which may be dry or humid) [86]

**Table 2 jpm-13-00944-t002:** The Rutherford classification.

Stage 0	Asymptomatic
Stage 1	Mild claudication
Stage 2	Moderate claudication—the distance that delineates mild, moderate, and severe claudication is not specified in the Rutherford classification, as it is in the Fontaine classification.
Stage 3	Severe claudication
Stage 4	Rest pain
Stage 5	Ischaemic ulceration not exceeding ulcers of the digits of the foot
Stage 6	Severe ischaemic ulcers or frank gangrene [86]

**Table 3 jpm-13-00944-t003:** Studies examining the relationship between peripheral arterial disease and mortality.

Study Name	Number of Patients	Trial Type	End Point
FOURIER trial	27,564	Prospective, randomized, double-blind, placebo-controlled trial	Evolocumab significantly reduced the risk of the primary composite end point of cardiovascular death, myocardial infarction, stroke, hospitalization for unstable angina, or coronary revascularization [127].
High mortality risks after major lower extremity amputation in Medicare patients with peripheral artery disease 2	186,338 patients with identified PAD who underwent major LE amputation	Retrospective study	Mortality rate 13.5% at 30 days, 48.3% at 1 year, and 70.9% at 3 years. Age per 5-year increase (hazard ratio [HR] 1.29, 95% CI 1.29–1.29), history of heart failure (HR 1.71, 95% CI 1.71–1.72), renal disease (HR 1.84. 95% CI 1.83–1.85), cancer (HR 1.71, 95% CI 1.70–1.72), and chronic obstructive pulmonary disease (HR 1.33, 95% CI, 1.32–1.33) were all independently associated with death after major LE amputation [140].
LIPAD study	331	Prospective study	Mortality rates at 10 years were 29% in non-diabetic PAD patients versus 14% in age- and sex-matched non-diabetic controls (risk ratio (RR), 2.31; 95% confidence interval (CI), 1.54–3.47; *p* < 0.001), and 58% in diabetic PAD patients versus 19% in age- and sex-matched diabetic controls (RR, 4.06; 95% CI, 2.67–6.18; *p* < 0.001) [139].
Peripheral Artery Disease of the Lower Limbs and Morbidity/Mortality in Type 2 Diabetics	269 type 2 diabetics, of which 63 had peripheral artery disease	Retrospective study	39 patients had died, of whom 19 had PAD in 1996 (30.1%) and 20 did not (9.7%) (*p* = 0.001).16 died in the group with an ABI of <0.9 (30.2%) and 21 (10.1%) in the group with normal ABI values (*p* = 0.001). Seven (13.2%) patients died due to a cardiovascular cause with a pathological ABI, and eight (3.9%) with a normal value (*p* = 0.009) [138].
EUCLID study	13,885	Prospective study, multicenter, randomized, double-blind	A total of 1263 out of 13,885 (9.1%) patients died (median follow-up: 30 months). There were 706 patients (55.9%) with a cardiovascular cause of death and 522 (41.3%) with a noncardiovascular cause of death [137].

## Data Availability

Not applicable.

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
