# Peer review of "Acute Coronary Syndrome: Disparities of Pathophysiology and Mortality with and without Peripheral Artery Disease"

_jpm, 2023, doi:10.3390/jpm13060944_

Round 1
Reviewer 1 Report
I read with interest the paper entitled “ Acute coronary syndrome: disparities of pathophysiology and mortality with and without peripheral artery disease” by Gherasie et al.
After careful reading I have the following comments and suggestions:
This is a well-written manuscript. However, the authors could be focused more on the topic of the review. More specifically, they could provide more data from published reports regarding the co-existence of coronary artery disease and peripheral vascular disease. Two important subpopulations are those patients who undergo percutaneous coronary intervention (PCI) for acute coronary syndrome (ACS) or coronary artery bypass grafting (CABG). It is of importance to discuss the outcomes of these procedures in the case of presence of peripheral artery disease (PAD). The inclusion of such data will add significant value to the current review. The authors are mentioning the PCI method in section 5 under the diagnostic approach.
Since the mortality and the adverse outcomes occur not only from the natural history and the progression of ACS and PAD, but also after medical, endovascular or surgical interventions these outcomes should be discussed accordingly in order to reveal possible disparities which may be related with the mode of therapeutic intervention.
Minor corrections:
1. Line 189: by foam cell formation characterized … it can be “cell formation which is the result of lipid …”
2. Line 203: Atherosclerosis plaques … correct to “Atherosclerotic plaques …”
3. Line 217: atherosclerosis plaque … correct to “atherosclerotic plaque …”
4. Line 234: cells (Tregs) has been … correct to “have been…”
5. Line 446: In general, ALI … the abbreviation ALI is not provided previously in the text
6. Line 547: is the key to decreasing … correct to “key to decrease …”
Author Response
Thank you for your recommendations. We corrected 1,2,3,4,6 as you suggested. The "ALI" appears for the first time on line 383 with the abbreviation.
In addition, we provided a conclusion regarding mortality and adverse outcomes associated with Acute Coronary Syndrome and Peripheral Artery Disease from medical, endovascular, or surgical interventions. There is a paragraph describing the outcomes of patients who had percutaneous coronary interventions for acute coronary syndrome or coronary artery bypass grafting in the presence of peripheral artery disease.
Reviewer 2 Report
The manuscript is comprehensive and well written. However, there are certain aspects that need to be modified:
-Тitle 3 mentions pathological aspects that are not described in the text that follows. It is necessary to either supplement or change the name of the chapter.
-Please change CD4 T to CD4+ T cells (ex. line 184 and 234). This needs to be checked throughout the whole manuscript.
-When organizing the manuscript, it is better to order the chapters in such a way that those related to pathological aspects appear first, followed by those related to diagnostics.
-A large number of references in this paper are older than 10 years, it is necessary to use a more recent reference where possible and reduce the number of references older than 10 years
-Technically, some sentences are bolded without a clear meaning of their emphasis. Also, the title of Table 3 and Conclusion need to be technically changed so that the title is above the table in the right place in the manuscript
Author Response
Thank you for your recommendations. We updated title 3, changed CD4 to CD4+, and updated the references. More than 50% are no older than 5 years.